# Selective Arterial Embolization with N-Butyl Cyanoacrylate Prior to CT-Guided Percutaneous Cryoablation of Kidney Malignancies: A Single-Center Experience

**DOI:** 10.3390/jcm10214986

**Published:** 2021-10-27

**Authors:** Olivier Lopez, Olivier Chevallier, Kévin Guillen, Pierre-Olivier Comby, Julie Pellegrinelli, Claire Tinel, Nicolas Falvo, Marco Midulla, Eric Mourey, Romaric Loffroy

**Affiliations:** 1Image-Guided Therapy Center, Department of Vascular and Interventional Radiology, François-Mitterrand University Hospital, 14 Rue Paul Gaffarel, BP 77908, 21079 Dijon, France; olivier-lopez@live.fr (O.L.); olivier.chevallier@chu-dijon.fr (O.C.); kguillen@hotmail.fr (K.G.); julie.pellegrinelli@chu-dijon.fr (J.P.); nicolas.falvo@chu-dijon.fr (N.F.); marco.midulla@chu-dijon.fr (M.M.); 2Imaging and Artificial Vision (ImViA) Laboratory-EA 7535, University of Bourgogne/Franche-Comté, 9 Avenue Alain Savary, BP 47870, 21078 Dijon, France; pierre-olivier.comby@chu-dijon.fr; 3Department of Neuroradiology and Emergency Radiology, François-Mitterrand University Hospital, 14 Rue Paul Gaffarel, BP 77908, 21079 Dijon, France; 4Department of Nephrology and Renal Transplantation, François-Mitterrand University Hospital, 14 Rue Paul Gaffarel, BP 77908, 21079 Dijon, France; claire.tinel@chu-dijon.fr; 5Department of Urology and Andrology, François-Mitterrand University Hospital, 14 Rue Paul Gaffarel, BP 77908, 21079 Dijon, France; eric.mourey@chu-dijon.fr

**Keywords:** embolization, cyanoacrylate, NBCA, lipiodol, ablation, cryoablation, kidney neoplasms, cryosurgery

## Abstract

The study’s purpose was to assess the safety, feasibility and efficiency of selective arterial embolization (SAE) using N-butyl cyanoacrylate (NBCA) glue before percutaneous cryoablation (PCA) of renal malignancies in patients whose tumor characteristics and/or comorbidities resulted in an unacceptable risk of bleeding. In this single-center retrospective study of 19 consecutive high-risk patients (median age, 74 years) with renal malignancies managed in 2017–2020 by SAE with NBCA followed by PCA, data about patients, tumor and procedures characteristics, complications, renal function and hemoglobin concentration before and after treatment, as well as recurrence were collected. Charlson comorbidity index was ≥4 in 89.5% of patients. Ten patients were treated by antiplatelet and/or anticoagulant therapy. Median tumor largest diameter was 3.75 cm (range, 1–6.5 cm) and R.E.N.A.L. nephrometry score was ≥7 in 80%, indicating substantial tumor complexity. No major complications were recorded and minor complications occurred in 7 patients. No residual tumor was found at 6-week imaging follow-up in 18/19 patients. Tumor recurrence was visible in 1/16 patients at 6-month imaging follow-up. No significant difference was found for renal function after treatment (*p* = 0.07), whereas significant decrease in hemoglobin concentration was noted (*p* = 0.00004), although it was relevant for only one patient who required only blood transfusion and no further intervention. SAE prior to PCA is safe and effective for managing renal malignancies in high-risk patients.

## 1. Introduction

Worldwide, kidney malignancies account for 2% of all cancers and are being increasingly diagnosed. In 2018, 63,000 new cases and 15,000 deaths due to kidney cancer occurred in the United States and about 350,000 new cases worldwide [1]. The increased use of cross-sectional imaging techniques as primary diagnostic tools has increased both the incidence of renal cancer and the proportion of tumors detected at an early stage, when nephron-sparing surgery or percutaneous ablation remain effective.

In recent decades, percutaneous ablative therapies such as microwave ablation (MWA), radiofrequency ablation (RFA) and cryoablation have become the reference standard treatment for renal malignancies in patients who are ineligible for conservative surgical resection [2,3,4,5,6,7]. These treatments are most effective for small tumors with exophytic growth [8]. Cryoablation consists in inducing local freeze-and-thaw cycles to cause mechanical destruction, osmotic lysis and immune-mediated apoptosis [9,10]. The main indication for percutaneous renal cancer ablation is the patient ineligibility for radical or partial nephrectomy.

Percutaneous cryoablation (PCA) performs better compared to heat-based ablation methods due to major heat-sink effect induced by the larger vessels of the kidney hilum. Other advantages include better conservation of urothelial tissue, feasibility under local anesthesia, the ability to use multiple probes simultaneously and the virtual absence of pain [11,12,13,14]. In addition, the ice ball that forms at the tip of the cryoprobe, where ablation occurs, can be readily monitored in real time using ultrasound [15], computed tomography (CT), or magnetic resonance imaging (MRI) [11]. In addition, the tissue destruction occurs according to a predictable pattern. Finally, the complication rate is low compared to heat-based ablation techniques and surgery [14,16,17]. However, cryoablation may induce more severe acute systemic effects, due to the lysis of cell membranes and has no cautery effect [18]. In patients with stage 1 renal cancer, PCA provided a 10-year disease-specific survival of 94%, similar to that observed after partial or radical nephrectomy, with higher overall survival, notably in patients with multiple comorbidities [19].

Selective arterial embolization (SAE) is a widely studied method for decreasing the vascularity and volume of renal malignancies, thereby facilitating subsequent complete surgical tumor excision and limiting intraoperative bleeding [20,21,22,23]. In a nested matched case-control study, survival in 118 patients who underwent renal artery embolization before nephrectomy for renal cell carcinoma was 62% and 47% at 5 and 10 years, respectively, compared with 35% and 23% in 116 patients without pre-operative embolization [22]. A study of SAE followed by partial nephrectomy for renal tumors suggested that renal function impairment was similar to that after other partial nephrectomy techniques [23].

The data on the use of SAE prior to percutaneous renal tumor ablation are scarcer. A literature review, that focused specifically on SAE performed before RFA of stage 1 renal cancer, concluded that the combination was feasible, highly effective and safe for T1a tumors in challenging locations and for T1b tumors [24]. For central and mixed tumors, better outcomes with SAE before ablation were also suggested by another literature review compared with ablation alone, although some studies included only small tumors [25]. Thus, further information on outcomes after SAE performed specifically before PCA of renal malignancies is needed.

The choice of the embolic agent for SAE prior to percutaneous ablation or surgery remains operator-dependant. Different embolic agents used alone or in combination have been used: iodized oil (Lipiodol^®^ Ultrafluid, Guerbet, Aulnay-sous-Bois, France) followed by gelatin sponge [26], ethiodol-ethanol mixture [27], ethanol mixed with iodized oil [28,29], polyvinyl alcohol (PVA) [28,29,30], microspheres with platinum coils [31], PVA in addition, to Lipiodol^®^ [32]. To our knowledge, no study reported the use of N-butyl cyanoacrylate glue (NBCA) in this particular setting. NBCA has been used for embolization of renal carcinoma in one study, mostly for palliative management [32] and its use has also been described for angiomyolipoma management [33,34,35].

Before administration, NBCA requires to be mixed with Lipiodol^®^ to make it radiopaque and to delay its polymerisation rate. The combination with Lipiodol^®^ also offers the potential benefit of enhancing tumor localization during CT-guided ablation [32,36,37], that could be particulalry useful in settings of poorly visualized endophytic renal tumor [37], or in patients with compromised renal function for whom contrast medium injection should be reduced.

The aim of this study was to evaluate the safety, feasibility and efficiency of SAE using NBCA as an embolic agent prior to PCA of renal malignancies in patients at high risk of bleeding complications who were not eligible for surgery.

## 2. Materials and Methods

### 2.1. Study Population

In this single-center retrospective study, consecutive patients not eligible for partial nephrectomy who underwent SAE followed by PCA for one or more localized renal malignancies between 2017 and 2020 at our institution, were included. Patients were eligible for study inclusion regardless of malignant tumor histology.

In accordance with French legislation, informed consent was not required for this retrospective study of anonymized data. The database was reported to the French data protection authority (Commission Nationale de l’Informatique et des Libertés [CNIL]).

For each patient, the decision to perform PCA was made at a multidisciplinary meeting, based on the characteristics of the malignancy and patient’s surgical risk. The reason for performing SAE before PCA was an increased risk of bleeding due to tumor hypervascularity and/or antiplatelet or anticoagulant therapy that could not be stopped and/or substantial coagulation disorders.

The patients were admitted on the day of SAE. PCA was performed on the next day. Patients were discharged home the day after PCA if free of complications and otherwise as soon as possible. As required by French law, written informed consent for each of the invasive procedures was obtained.

The following data regarding patients and tumors characteristics were recorded: age; gender; presence of 1 or 2 kidneys; patients Charlson Comorbidity Index (CCI), that was defined as low for a CCI between 1 and 3, high for a CCI between 4 and 6 and very high for a CCI ≥ 7 (Appendix A, Table A1) [38]; antiplatelet medication; anticoagulant therapy; tumor’s largest diameter (≤4 cm, 4 < size ≤ 6 cm, >6 cm); tumor’s histology; tumor’s complexity score as defined by the R.E.N.A.L. nephrometry score (Appendix A, Table A2) [39], which has been shown to predict ablation efficacy and complications [40,41,42] and was used to classify the tumors as exhibiting low, intermediate, or high complexity.

### 2.2. Selective Arterial Embolization

All procedures were performed by one experienced interventional radiologist (R.L.), using a Philips Allura Xper FD 20 angio room (Philips, Best, The Netherlands). After local anesthesia and common femoral artery puncture under ultrasound guidance, transfemoral selective renal angiography was performed through a 6 Fr sheath using 5 Fr diagnostic catheters. Tumor and its supplying vessels were identified. A microcatheter (Progreat^®^, Terumo Interventional Systems, Tokyo, Japan), ranging from 2.0 Fr to 2.7 Fr, was then positioned as selectively as possible in the feeding artery to spare as much healthy parenchyma as possible. If necessary, cone-beam CT was performed to identify the feeding arteries and to assess the perfused volume.

Once the appropriate positioning had been confirmed, the microcatheter dead space was filled with an anionic solution, dextrose 5%, to avoid intracatheter glue polymerization and embolization was performed under strict fluoroscopic control with N-butyl cyanoacrylate glue (Glubran^®^2, GEM, Viareggio, Italy) mixed with Lipiodol^®^. This embolic agent has been proven effective for tumor devascularization [34,35]. The glue/Lipiodol^®^ mixture ratio was at the discretion of the interventional radiologist and ranged from 1/3 to 1/6. Great care was taken to avoid any reflux and nontarget embolization. Embolization was repeated the same way in other arteries feeding the tumor that had been previously identified. SAE was considered successful when occlusion of at least one feeding artery was achieved with only partial or no tumor blush upon subsequent angiography and complete SAE was defined as occlusion of all feeding arteries with no residual tumor blush upon subsequent angiography [43]. Partial tumor blush was mainly due to small feeding arteries either not eligible to microcatheterization or impossible to embolize while preserving adjacent healthy kidney parenchyma. The partial residual tumor blush was always less than 50% of the tumor volume on the postembolization angiogram. The presence of partial tumor blush remained acceptable if all major feeding arteries were occluded.

The femoral access was closed using the FemoSeal^®^ device (Terumo, Shibuya City, Tokyo, Japan). After the procedure, analgesics were given on demand. The embolization procedure time and fluoroscopy time were recorded, as well as the amount of contrast agent used and the ionizing radiation exposure expressed as the dose-surface product (DSP) in Gy·cm^2^.

### 2.3. Percutaneous Cryoablation Procedure

PCA procedure was performed under CT guidance (64-slice helical CT scanner, Siemens SOMATOM Definition AS, Siemens Healthcare, Erlangen, Germany) using argon gas-based cryostat system (VISUAL ICE Cryoablation System, Galil Medical Inc., Boston Scientific, Marlborough, MA, USA) and 17-gauge 1.5 IceRod™ CX cryoablation needles (Galil Medical Inc., Boston Scientific), by the same interventional radiologist who performed SAE. The type of anesthesia, general or local, was decided jointly by the interventional radiologist and anesthesiologist. Patient position was governed by the location of the tumor and patient comfort or, when general anesthesia was performed, the anesthesiologist’s choice. A preliminary unenhanced or enhanced CT-scan was obtained for target lesion visualization and cryoprobes placement planification. The use of contrast injection was at the discretion of the interventional radiologist, since the Lipiodol^®^ uptake in the tumor area after SAE often allowed an adequate visualization of the tumor on the unenhanced CT-scan images [32,36,37]. When tumor histology had not yet been determined, a percutaneous biopsy was performed using the coaxial technique with an automatic or semiautomatic 18-gauge biopsy needle through a 17-gauge coaxial needle. Multiple scans were acquired during cryoprobe positioning and multiplanar and volumetric reconstructions were obtained when needed. Additional probes were placed if coverage was insufficient. The number of cryoprobes varied with the size and shape of the tumor. When necessary, the hydrodissection technique using glucose 30% solution was used to displace and spare the surrounding critical structures. The protocol included a 10 min freeze, an 8 min thaw (6 min passive then 2 min active), a 10 min refreeze and a 1 to 2 min active thaw [44,45]. Unenhanced CT images were acquired during ablation to assess tumor coverage by the ice ball [11]. Complete ablation was considered achieved if the cryoablation zone included the entire tumor and safety margins of 5 mm or more, accordingly to protocol [46]. The use of cryoprobes cautery function for track ablation while probes were withdrawn was at the discretion of the interventional radiologist. At the end of the procedure, control unenhanced CT-scan was performed to look for immediate complications.

### 2.4. Oncologic Imaging Follow-Up

Follow-up renal MRI was performed at 6 weeks to look for residual tumor and for evidence of delayed complications [46,47,48] and then at 6 months to look for local tumor recurrence [49]. Technique efficacy at 6-week and 6-month were defined as absence of residual tumor on the 6-week MRI and absence of local tumor recurrence on the 6-month MRI, respectively [46]. When patients underwent other imaging after 6 months, MRI or CT scan, the last exam was kept to establish our median follow-up.

### 2.5. Safety Evaluation

Treatment-related complications occurring within 30 days of the PCA procedure were recorded using the revised Clavien–Dindo classification [50,51]. Complications grade III or higher were considered major. Other complications were considered clinically significant if they required specific treatment (grade II) and/or extended the hospital stay length and/or required readmission within 30 days after the procedure.

Pain was evaluated after each procedure and before hospital discharge, using a numerical rating scale (NRS) ranging from 0 (no pain) to 10 (worst pain imaginable) [52].

Doppler ultrasonography was systematically performed to check the puncture site on the day after cryoablation (i.e., on the day of discharge in patients without complications).

The estimated glomerular filtration rate (eGFR), that was calculated using the Modification of Diet in Renal Disease (MDRD) equation in mL/min/1.73 m^2^ and the blood hemoglobin concentration (g/dL) were both assessed before SAE and after PCA procedure.

### 2.6. Statistical Analysis

Normality was assessed for continuous variables. Discrete variables and continuous variables, for which normality was rejected, were described using the median and the range values (minimum-maximum). Continuous variables, for which normality was accepted, were described using mean and standard deviation (SD) and the variables comparison was carried out with a paired Student t-test. Statistical analyses were performed using Stata 14.0 software (StataCorp, College Station, TX, USA).

## 3. Results

### 3.1. Patients and Tumors Characteristics

Nineteen patients (median age, 76 years old; 17 males) with 20 tumors were identified and included. Table 1 and Table 2 report the main characteristics of the patients and the tumors, respectively. Among the 17 patients (85.9%) who had a CCI greater or equal to 4, 8 patients (42.1%) had a score greater or equal to 7. Antiplatelet medication and/or anticoagulant therapy were taken by 10 patients (52.6%).

Nine out of 19 patients (47.4%) had a tumor larger than 4 cm. One patient presented two tumors: one of these lesions was a recurrence after a previous PCA without SAE, the other one was a new lesion. None of the other patients had history of previously treated renal malignancy. For the 20 lesions, the R.E.N.A.L. nephrometry score ranged from 4 a, which is considered the most technically favorable, to 10 p, the least technically favorable score being 12 ph [39]. Most lesions had scores of 7 or more, indicating substantial tumor complexity. Among 20 tumors, 16 (80%) were renal clear-cell carcinoma, 2 were chromophobe carcinoma and 2 were undetermined malignancies. For the latter two, decision to perform ablation therapy was made at multidisciplinary meetings in other oncologic centers. Patients were then addressed to our center for PCA. Multidisciplinary meetings reports considered the diagnosis of renal clear-cell carcinoma but histological reports were not provided. However, both tumors demonstrated typical imaging findings of clear-cell carcinoma.

### 3.2. Selective Arterial Embolization Intervention

Successful SAE was achieved for all patients, but complete SAE in 11 patients (57.9%). Complete SAE could not be achieved in the remaining eight patients since it would have led to an unacceptable loss of surrounding healthy tissue. However, for cases with incomplete embolization, devascularization was obtained for most of the tumor volume, always greater than 50% on control angiogram. The most used NBCA/Lipiodol^®^ ratio range was 1/3 to 1/5, in 18 patients. Mean procedure time was 93 ± 43 min and mean fluoroscopy time was 18 ± 11 min. The mean DSP for the SAE procedure was 120.7 ± 68.7 Gy.cm^2^. The mean volume of injected contrast medium was 119 ± 50 mL. The mean NRS pain score was 0.89 ± 1.52 after SAE. Table 3 lists the main characteristics of the SAE procedure. Figure 1 and Figure 2 show examples of SAE of renal cell carcinoma before PCA.

### 3.3. Percutaneous Cryoablation Procedure

Table 4 reports the features of the PCA procedures. Complete ablation was obtained for all patients and no immediate complications were visible on the control unenhanced CT-scan performed at the end of the procedure. Five patients required hydrodissection prior to ablation. No patient reported more than moderate pain after PCA. No ice-ball cracking occurred. Figure 3 and Figure 4 show examples of PCA of renal cell carcinoma after SAE.

### 3.4. Oncologic Outcomes

All the 19 patients underwent 6-week follow-up MRI. Only one patient had residual tumor visible at 6 weeks. Therefore, technique efficacy rate at 6-week was 94.7% (18/19 patients). The patient with residual tumor was the one with the largest tumor (6.5 cm) in the patient series. The tumor’s R.E.N.A.L. nephrometry score was 10 a. A new cryoablation was then successfully performed.

Of the 19 patients, 3 were then lost to follow-up because of decompensation of Alzheimer’s disease (*n* = 1) or due to the COVID-19 pandemic (*n* = 2). Of the remaining 16 patients with data available for 6 months follow-up analysis, MRI at 6 months showed tumor recurrence in 1 (6.3%) patient. The R.E.N.A.L. nephrometry score of this lesion was 10 p and the lesion size was 5.2 cm. Technique efficacy rate at 6-month was 87.5% (14/16 patients). A second PCA procedure was thus performed for this patient with no residual tumor visible on the second 6-week follow-up MRI.

Median follow-up was 14 months (range, 2–41). After PCA, follow-up imaging data for the periods 6–12 months, 12–18 months and after 18 months, were available for 84.2% (16/19 patients), 52.6% (10/19 patients) and 42.1% (8/19 patients) patients, respectively. At last follow-up, characterized by the time of the last imaging of the renal tumor available (CT-scan or MRI) or the date of death from another cause, no other patient had evidence of tumor recurrence.

### 3.5. Safety

No major 30-day complications were recorded. An adverse event requiring an extension of the hospital stay occurred in 3 (16%) patients: Two patients experienced urinary tract infection. One patient required a red-blood-cell transfusion due to a significant hematoma, although he was not treated with antiplatelet medication or anticoagulant therapy. No additional intervention was required for hemostasis. Concerning the other minor complications, pneumothorax occurred in 1 patient during the PCA that did not required chest tube placement. Small peri-renal hematoma was found in 3 patients, among whom only one was treated with anticoagulant therapy, the two others were not treated by antiplatelet medication nor anticoagulant therapy. No complications related to the SAE occurred. No complications regarding the femoral puncture site were noted during the systematic control by Doppler ultrasound.

One significant dilatation of the pelvicalyceal system was found in 1 patient on the 6-week follow-up MRI but was asymptomatic without acute renal failure and resolved spontaneously without the need for further intervention. No other delayed complication at 6 weeks was observed.

There was no significant difference in the eGFR between the pre-procedure and post-procedure values: the mean eGFR was 76.7 ± 26.8 mL/min/1.73 m^2^ before the procedures and 72.9 ± 28.0 mL/min/1.73 m^2^ after the procedures (*p* = 0.07). The mean hemoglobin concentration value was 13.3 ± 2.2 g/dL before the procedures and 12.3 ± 2.2 g/dL after the procedures, with a significant difference (*p* = 0.00004), although no patient presented clinical sign of anemia except the one with the significant hematoma. The latter was the only one that required a blood transfusion.

Pain requiring step 2 or higher analgesic during the hospital stay occurred in 11 patients (57.9%). None of the patients reported any pain at hospital discharge. Mean hospital stay length was 2.4 ± 0.9 days.

Table 5 reports the study results related to safety.

## 4. Discussion

In this study, that included 19 patients who were not eligible for surgery and were referred for PCA of kidney malignancies because of high risk for bleeding, combination of SAE using NBCA followed with PCA was safe, feasible and efficient. No major complication occurred, although many of the patients were elderly and suffered from multiple comorbidities. None of the patients reported any pain at hospital discharge, with a short mean hospital stay length of 2.4 ± 0.9 days. Residual tumor at 6-week follow-up was only seen in the patient who had the largest tumor (6.5 cm) and local recurrence at 6-month follow-up was found in only one patient.

Percutaneous ablation techniques allow curative treatment of kidney malignancies in patients who are ineligible for conservative surgical resection [2,3,4,5,6,7]. A meta-analysis suggested that for ablation of small renal masses, PCA resulted in fewer retreatments and improved local tumor control and may be associated with a lower risk of metastatic progression compared with RFA [11]. In a prospective nonrandomized study including 120 renal tumors in 95 patients treated with PCA because their condition did not allow surgery, only five grade II complications and four grade III–V complications occurred [53]. However, six of them were related to bleeding complications, with four large perirenal hematomas, two of which requiring blood transfusion, one case of hematuria which required transient ureteral stent placement, one compressive subscapular hematoma on a single kidney that caused anuria and urgent surgical decompression [53]. With 101 percutaneous, 52 laparoscopic and 9 open surgery procedures included in Sidana et al.’s retrospective study, hematoma was the second most common complication that occurred in 10 procedures (6.2%) [18]. In this study, univariate regression analysis showed that mass size, number of cryoprobes and chronic anticoagulation, were significantly associated with an increased hematoma incidence, whereas with multivariate logistic analysis the total number of cryoprobes was the only significant predictor of hematoma [18]. However, they found significant correlations between mass size and total number of probes and between the total number of probes and anticoagulants [18]. In a retrospective review of all immediate and delayed complications of PCA of renal tumors <7 cm from a multicenter database, with 713 renal tumors ablated in 647 procedures, the most frequent complication was bleeding with 21 cases (3.2%), of which 9 cases required subsequent treatment [17]. Although significant bleeding events incidence after PCA appears to be low, these complications remain a major concern in fragile patients, for whom PCA is the recommended curative treatment. In addition, antiplatelet medication and anticoagulation therapy are increasingly prescribed and often cannot be stopped for the procedure because of cardiovascular comorbidities. The authors proposed SAE before kidney tumor intervention to reduce the bleeding risk and improve oncologic outcomes [5,20]. With a total of 21 lesions in 19 patients, in Miller et al.’s study, 17 lesions were treated with PCA alone and 4 with SAE and PCA [5]. Nine bleeding complications (hemorrhage, hematuria or hematoma) occurred in the group treated with PCA only, whereas no bleeding complication was reported in the group treated with both SAE and PCA [5]. Embolization significantly decreased complications between size-matched lesions without impacting renal function or recurrence [5]. This is in contrast with another retrospective study that included 9 patients treated with SAE and PCA and 18 patients treated with PCA alone, in which no significant difference was found in the rate of complications [43]. However, a comparison of preprocedural and postprocedural hematocrit showed a 2.1 ± 15.4% decrease in the SAE + PCA group versus 9.3 ± 8.3% in the PCA alone group, although the difference did not reach statistical significance (*p* = 0.152) [43]. In our study, although a notable proportion of patients were treated with antiplatelet or anticoagulant therapy, only four bleeding events occurred with three small peri-renal hematomas and one larger hematoma, the latter being the only one that required blood transfusion. Furthermore, nine lesions were larger than 4 cm, which is the most commonly accepted size limit for cryoablation. It has been shown that the risk of bleeding increases with the tumor size [54]. In addition, 16 lesions had R.E.N.A.L. nephrometry score ≥7, indicating substantial tumor complexity and higher risk of bleeding [39,40,41]. Similarly, the number of cryoprobes used for ablation, that has been shown to correlate with bleeding risk [18], was quite high in our study, with a median of 7. Although, the mean hemoglobin concentration value was significantly lower after the procedures, no patient presented clinical signs of anemia except the one with the significant hematoma. This is consistent with Gunn et al. study’s results, in which 3 perinephretic hematomas occurred with 10 PCA procedures performed after SAE [36]. However, more data are required to know the real impact of SAE performed prior to PCA on the bleeding risk.

In addition, and in agreement with other reports, no significant change in eGFR after the procedures was found in our study, suggesting that SAE before PCA is safe regarding the renal function [5,36,43]. In addition, no complication related to SAE, such as puncture site complication, occurred in our study. Patient’s pain following SAE was at most moderate.

The patient follow-up time in our study was quite short and therefore did not allow to draw definite conclusions in terms of oncologic outcomes. However, the technique efficacies at 6-week and 6-month were very high (94.7% and 87.5%, respectively), despite the proportion of large tumors included. Among the 19 patients included in this study, only one had residual tumor at 6-week follow-up and among the 16 patients with available data at 6 months, only 1 recurrence occurred. Both patients had quite large and complex lesions. The first had the largest tumor in the series (6.5 cm), with a tumor R.E.N.A.L. nephrometry score of 10 a and the second had a tumor of 5.2 cm with a R.E.N.A.L. score of 10 b. It is well known that the larger the tumor, the higher the risk of residual disease and recurrence. In a database study, tumor size greater than 3 cm was associated with significantly higher cancer-specific mortality at 5 years even after propensity-score matching (hazard ratio, 2.86, *p* < 0.001), compared with size ≤ 3 cm [54]. PCA alone for either T1a and T1b renal malignancies demonstrated technical success of 94% after a single ablation and a primary technical success rates of 94% after one year of follow-up in a study that included 95 patients with 120 renal tumors [53].

To our knowledge, the present study is the first to explore the use of NBCA glue in this indication. Whereas the use of NBCA has widely been reported in acute bleeding setting to achieve hemostasis [55,56], few papers reported its use for tumor devascularization [34,35]. NBCA offers several advantages. Due to its liquid nature, the operator can inject NBCA through a flow-directed strategy that allows distal embolization, particularly of small or tortuous arteries, such as observed with tumor neovasculature. The viscosity of the liquid can be chosen by adjusting the glue/Lipiodol^®^ ratio. The mixture can show sufficient low viscosity to allow distal embolization of the tumor feeding arteries. In addition, its quick administration and rapid polymerization allow fast embolization, decreasing the procedure time. Lipiodol^®^ makes the NBCA/lipiodol mixture radiopaque, allowing for easier control under fluoroscopy, compared with other embolic materials which are not directly visualized such as particles [34]. In addition, Lipiodol^®^ allows to contrast the tumor on unenhanced CT-images, enabling reduction of targeting inaccuracies [32,36,37]. Furthermore, additional contrast medium injection may thus be avoided, particularly in patients with compromised renal function, while allowing a good visualization of the tumor during PCA. The NBCA used as an embolic agent in our study was Glubran^®^2. It has advantages over classical cyanoacrylate glues such as Histoacryl^®^. First, the polymerization reaction is slower with Glubran^®^2, thus making handling and release easier and allowing a more distal embolization. In Glubran^®^2, another monomer, metacryloxysulfolane, is added to the NBCA, producing a more pliable and stable polymer whose milder exothermic reaction (45 °C) results in less inflammation and histotoxicity than is the case with Histoacryl^®^ [57,58].

Our study had several limitations. First, this was a single center study with retrospective design. Second, the sample size was small and there was a bias of selection since only high-risk bleeding patients were treated by SAE prior to PCA. Although studies reported PCA alone outcomes with much larger population, previously published case series that focused on SAE before PCA had number of patients similar to our study. Third, follow-up time was quite short, especially for renal tumors that are usually slow growing and some patients were lost to follow-up because of comorbidity or the COVID-19 pandemic. Therefore, no definite conclusions could be drawn regarding oncologic outcomes. Fourth, histology of two tumors was unknown. However, imaging findings and their growth were highly suggestive of renal cell carcinoma. Last, no comparative group without any increased risk of bleeding due to tumor hypervascularity and/or antiplatelet or anticoagulant therapy that could not be stopped and/or substantial coagulation disorders was assessed. Further comparative studies are needed.

## 5. Conclusions

The present study demonstrated that SAE using NBCA as an embolic agent prior to PCA for malignant kidney tumors is safe, feasible and efficient in patients who were not eligible for surgery and at high risk for bleeding. Randomized trials are warranted to compare embolic agents in this indication and determine the benefits of SAE before PCA.

## Figures and Tables

**Figure 1 jcm-10-04986-f001:**
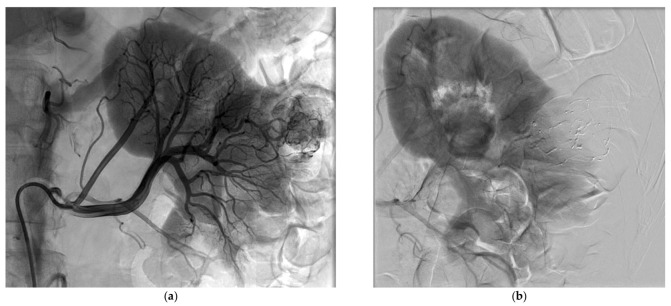
Selective arterial embolization (SAE) procedure using NBCA/Lipiodol^®^ mixture performed before percutaneous cryoablation of a renal cell carcinoma of the left kidney. (**a**) Angiogram showed hypervascularization of the renal malignancy with one feeding artery. (**b**) Control angiogram after SAE using NBCA/Lipiodol^®^ mixture (1/5 ratio) showing the complete occlusion of the tumor vessels and devascularization of the tumor whereas the surrounding healthy tissue was spared.

**Figure 2 jcm-10-04986-f002:**
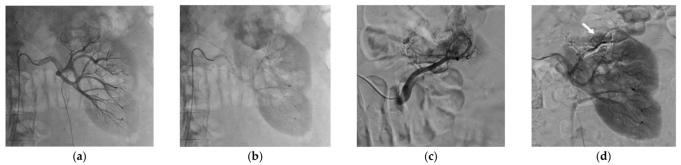
Selective arterial embolization (SAE) procedure using NBCA/Lipiodol^®^ mixture performed before percutaneous cryoablation of a renal cell carcinoma of the upper pole of the left kidney. (**a**,**b**) Angiogram showed hypervascularrenal malignancy. (**c**) Selective angiogram after embolization of a feeding artery of the tumor showing another feeding artery with residual tumor blush. (**d**) Control angiogram showing the occlusion of the two major feeding arteries after embolization with a small residual tumor blush (white arrow).

**Figure 3 jcm-10-04986-f003:**
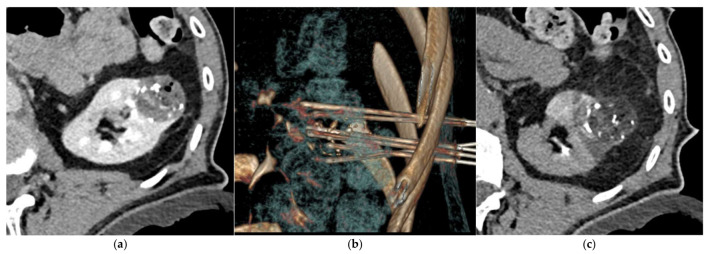
CT-guided percutaneous cryoablation procedure of a renal cell carcinoma of the left kidney performed after selective arterial embolization using NBCA/Lipiodol^®^ mixture. (**a**) Enhanced CT-scan for trajectory planification showing a complete devascularization of the tumor. Note the Lipiodol^®^ uptake in the tumor area that can increase targeting accuracy. (**b**) Volume rendering reconstruction showing the cryoprobes placement. (**c**) Unenhanced control CT-scan post-ablation showing the optimal coverage of the tumor without any immediate complication.

**Figure 4 jcm-10-04986-f004:**
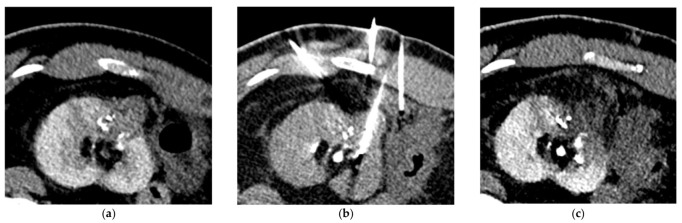
CT-guided percutaneous cryoablation procedure of a renal cell carcinoma of the left kidney performed after selective arterial embolization using NBCA/Lipiodol^®^ mixture. (**a**) Enhanced CT-scan for trajectory planification showing a complete devascularization of the tumor with lipiodol uptake. Note that the left colon is really close to the kidney. (**b**) CT acquisition showing the hydrodissection technique with needle placement for injection of dextrose solution to repel the colon and protect it. (**c**) Unenhanced control CT-scan post-ablation showing the optimal coverage of the tumor without any immediate complication.

**Table 1 jcm-10-04986-t001:** Characteristics of the 19 patients included in the study.

Patient Characteristics	Median (Range) or No. (%)
Age, years	76 (53–91)
Males/Females	17 (89.4)/2 (10.6)
Patients with two kidneys	19 (100)
Charlson comorbidity index (CCI)	
1–3	2 (10.5)
4–6	9 (47.4)
≥7	8 (42.1)
Antiplatelet medication *	
1 drug	3 (15.8)
2 drugs	1 (5.3)
Anticoagulant therapy **	5 (26.3)
Both antiplatelet and anticoagulant therapies	1 (5.3)

No., number; *, antiplatelet medication only, without anticoagulant therapy; **, anticoagulant therapy only, without antiplatelet medication.

**Table 2 jcm-10-04986-t002:** Characteristics of the 20 malignant kidney tumors in the 19 patients included in the study.

Tumor Characteristics	Median (Range) or No. (%)
Largest diameter, cm	3.75 (1–6.5)
≤4	11 (55)
4 < size ≤ 6	8 (40)
>6	1 (5)
R.E.N.A.L. nephrometry score	
4–6	4 (20)
7–9	13 (65)
10–12	3 (15)
Histology	
Renal clear-cell carcinoma	16 (80)
Other	4 (20)

No., number.

**Table 3 jcm-10-04986-t003:** Characteristics of the selective arterial embolization (SAE) procedures of the 20 malignant kidney tumors performed prior to percutaneous cryoablation in 19 patients.

SAE Characteristics	No. (%) or Mean ± SD
NBCA/Lipiodol^®^ ratio	
1:3	5 (26.3)
1:4	6 (31.6)
1:5	7 (36.8)
1:6	1 (5.3)
Procedure time, min	93 ± 43
Fluoroscopy time, min	18 ± 11
Contrast medium, mL	119 ± 50
Successful SAE *	19 (100)
Complete SAE **	11 (57.9)
Pain following SAE	
NRS = 0	13 (68.4)
NRS < 5	6 (31.6)
NRS ≥ 5	0 (0.0)

No., number; NBCA, N-butyl cyanoacrylate; SAE, selective arterial embolization; SD, standard deviation; NRS, numerical rating scale for pain ranging from 0 (no pain) to 10 (worst pain imaginable); *, successful SAE was achieved when occlusion of at least one feeding artery was achieved with only partial or no tumor blush upon subsequent angiography; **, complete SAE was defined as occlusion of all feeding arteries with no tumor blush upon subsequent angiography; min, minute; mL, milliliter.

**Table 4 jcm-10-04986-t004:** Characteristics of the percutaneous cryoablation (PCA) procedure of the 20 malignant kidney tumors performed after selective arterial embolization in the 19 patients.

PCA Characteristics	No. (%) *
No. of cryoprobes, median (range)	7 (4–10)
PCA procedure time, min (mean ± SD)	151 ± 58
Type of anesthesia	
General	14 (73.7)
Local	5 (26.3)
Complete ablation **	19 (100)
Post-cryoablation pain	
NRS = 0	9 (47.4)
NRS < 5	10 (52.6)
NRS ≥ 5	0 (0.0)

No., number; PCA, percutaneous cryoablation; min, minute; SD, standard deviation; NRS, numerical rating scale for pain ranging from 0 (no pain) to 10 (worst pain imaginable); *, variables are presented using No. (%) unless otherwise specified; **, complete ablation was considered achieved if the cryoablation zone included the entire tumor and safety margins of 5 mm or more.

**Table 5 jcm-10-04986-t005:** Safety of selective arterial embolization prior to percutaneous cryotherapy in 19 patients: Complications, pain at discharge and analgesic therapy.

Complications and Pain Management	No. (%)
30-day overall complications	7 (36.8)
Major *	0 (0)
Minor **	7 (36.8)
Urinary tract infection	2 (28.6)
Small peri-renal hematoma	3 (42.9)
Hematoma requiring blood transfusion	1 (14.3)
Pneumothorax that did not require chest tube placement	1 (14.3)
Puncture site complication	0 (0)
Analgesic step 2 or 3 therapy	11 (57.9)
Step 2	2 (10.5)
Step 3	9 (47.4)
NRS for pain at discharge = 0	19 (100)

No., number; *, defined as score ≥ Clavien–Dindo III; **, defined as score of Clavien–Dindo I or II.

## Data Availability

The data presented in this study are available on request from the corresponding author. The data are not publicly available due to identity reasons.

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
