# Peer review of "Selective Arterial Embolization with N-Butyl Cyanoacrylate Prior to CT-Guided Percutaneous Cryoablation of Kidney Malignancies: A Single-Center Experience"

_jcm, 2021, doi:10.3390/jcm10214986_

Round 1
Reviewer 1 Report
Thank you for giving me a chance to review this paper.
The authors described SAE with NBCA prior to PCA.
The topic of this manuscript is interesting. The manuscript is well written and easily readable, contained something new.
My specific comments are followings.
Title: OK
Abstract: OK
Keywords: OK
Introduction:
The introduction provides sufficient background, though, it is too long. It should be a little compact.
M & M:
p4, l-157. Authors described partial tumor blush upon subsequent angiography. To what extent is it acceptable? Please explain it.
p4, l-177. When tumor histology was yet unknown, a biopsy was performed after SAE.
Did SAE affect the result of histology of the tumor? 2 tumors were undetermined, though, weren’t they due to the biopsy after SAE?
p4, l-183. Authors described the hydrodissection technique. Please add how many cases you did in the result.
Results:
p6, l-247. For cases with incomplete embolization, how many percent did tumor enhancement remain in the angiography?
Discussion:
p9, l-340. It was a mistake at 6-week follow-up.
Was there no case with ice-ball cracking? Were 4 bleeding events due to ice-ball cracking or other causes such as bleeding tendency or multiple percutaneous punctures? I think that it was a good result if the bleeding risk decreased after SAE even though ice-ball cracking occurred.
Conclusion: OK
Figures: OK
References: OK
Author Response
REPLIES TO REVIEWER 1 COMMENTS
Thank you for giving me a chance to review this paper.
The authors described SAE with NBCA prior to PCA.
The topic of this manuscript is interesting. The manuscript is well written and easily readable, contained something new.
Authors: Thank you very much for your comment.
My specific comments are followings.
Title: OK
Authors: Thank you very much for your comment. The title has been shortened to answer to Reviewer 2 comment.
Abstract: OK
Authors: Thank you very much for your comment.
Keywords: OK
Authors: Thank you very much for your comment.
Introduction:
The introduction provides sufficient background, though, it is too long. It should be a little compact.
Authors: Thank you very much for your comment.The introduction has been shortened a bit as suggested.
M & M:
p4, l-157. Authors described partial tumor blush upon subsequent angiography. To what extent is it acceptable? Please explain it.
Authors: Thank you very much for your comment.It is quite difficult to assess accurately the volume ratio between non-embolized and embolized area. Also, we tried to avoid performing CBCT after embolization in order to minimize the amount of contrast injected and to preserve kidney function as much as possible. Furthermore, due to tumor heterogeneity with necrosis component, enhanced versus unenhanced tumoral volumes could not be assessed precisely. The evaluation was therefore subjective using angiography alone. Partial tumor blush was mainly due to small feeding arteries either not eligible to micro-catheterization or impossible to embolize while preserving adjacent healthy kidney parenchyma. The partial residual tumor blush was always less than 50% of the tumor volume on the postembolization angiogram. We believe that presence of partial tumor blush remained acceptable if all major feeding arteries have been occluded. It has been described and explained in the materials and methods section. A new figure was added to illustrate partial tumor blush.
p4, l-177. When tumor histology was yet unknown, a biopsy was performed after SAE.
Did SAE affect the result of histology of the tumor? 2 tumors were undetermined, though, weren’t they due to the biopsy after SAE?
Authors: Thank you very much for your comment.SAE does not affect histology of the tumor. We can perform it after SAE without any issue. It has been demonstrated in the literature. For these two specific tumors, no biopsy was performed. Patients came from other oncologic centers. Multidisciplinary meetings stated that tumors were CCR, but we failed to get the histology reports from the centers. However, both tumors demonstrated typical imaging findings of CCR. The following sentences have been added to the manuscript in the first paragraph of the result section: “For the latter two, decision to perform ablation therapy was made at multidisciplinary meetings in other oncologic centers. Patients were then addressed to our center for PCA. Multidisciplinary meetings reports considered the diagnosis of renal clear-cell carcinoma but histological reports were not provided. However, both tumors demonstrated typical imaging findings of clear-cell carcinoma.”
p4, l-183. Authors described the hydrodissection technique. Please add how many cases you did in the result.
Authors: Thank you very much for your comment.Hydrodissection techniques were required in 5 cases. It has been added in the text in the 3.3 paragraph of the results section. Furthermore, a new figure illustrating hydrodissection has been added.
Results:
p6, l-247. For cases with incomplete embolization, how many percent did tumor enhancement remain in the angiography?
Author: Thank you very much for your comment.For patientswith incomplete embolization, the remaining tumor enhancement was variable, always less than 50%, due to small additional feeding arteries. It has been added in the 3.2 paragraph of the results section.
Discussion:
p9, l-340. It was a mistake at 6-week follow-up.
Author: Thank you very much for your comment.We missed this mistake during our review process. The correction has been made.
Was there no case with ice-ball cracking? Were 4 bleeding events due to ice-ball cracking or other causes such as bleeding tendency or multiple percutaneous punctures? I think that it was a good result if the bleeding risk decreased after SAE even though ice-ball crackingoccurred.
Author: Thank you very much for your comment.No ice-ball cracking occurred in our series. We cannot conclude that bleeding risk decreased after SAE specifically in case of ice-ball cracking. However, we believe that SAE could indeed prevent bleeding in this setting.
Conclusion: OK
Authors: Thank you very much for your comment.
Figures: OK
Authors: Thank you very much for your comment.
References: OK
Authors: Thank you very much for your comment.
Reviewer 2 Report
I find the article very interesting and well written.
Title: try if possible to shorten the title a little. I do not think it is necessary to write computed tomography in full, CT is an acronym of international use.
Introduction: line 52 add the following bibliographic entry:
Carrafiello G, Mangini M, Fontana F, Recaldini C, Piacentino F, Pellegrino C, Laganà D, Cuffari S, Marconi A, Fugazzola C. Single-antenna microwave ablation under contrast-enhanced ultrasound guidance for treatment of small renal cell carcinoma: preliminary experience. Cardiovasc Intervent Radiol. 2010 Apr;33(2):367-74. doi: 10.1007/s00270-009-9745-x. Epub 2009 Nov 14. PMID: 19915901.
Fig. 2 If possible, I would put a post CT image, in the same phase of the pre CT; it seems a little late and the opacification of the cortical is badly highlighted.
If possible, I would add an additional case in the figure as the topic is really compelling.
Author Response
REPLIES TO REVIEWER 2 COMMENTS
I find the article very interesting and well written.
Authors: Thank you very much for your comment. This is a topic of great interest for our IR team.
Title: try if possible to shorten the title a little. I do not think it is necessary to write computed tomography in full, CT is an acronym of international use.
Authors: Thank you very much for your comment. The title has been shortened as suggested with acronym use for CT.
Introduction: line 52 add the following bibliographic entry:
Carrafiello G, Mangini M, Fontana F, Recaldini C, Piacentino F, Pellegrino C, Laganà D, Cuffari S, Marconi A, Fugazzola C. Single-antenna microwave ablation under contrast-enhanced ultrasound guidance for treatment of small renal cell carcinoma: preliminary experience. Cardiovasc Intervent Radiol. 2010 Apr;33(2):367-74. doi: 10.1007/s00270-009-9745-x. Epub 2009 Nov 14. PMID: 19915901.
Authors: Thank you very much for your comment. This reference has been added. Citations have been then renumbered and updated.
Fig. 2 If possible, I would put a post CT image, in the same phase of the pre CT; it seems a little late and the opacification of the cortical is badly highlighted.
If possible, I would add an additional case in the figure as the topic is really compelling.
Authors: Thank you very much for your comment. In order to preserve kidney function, we tried to minimize the amount of contrast injected. If no immediate complication was found on the post-procedure unenhanced-CT, no contrast injection was performed. Indeed, no immediate complications were reported for all patients. Therefore, we do not have post-procedure enhanced CT scan. However, new figures (3 and 4) of an additional case have been added as suggested.